# Two-component anomalous Hall effect in a magnetically doped topological insulator

Nan Liu[1,2], Jing Teng[1,2,3] & Yongqing Li [1,2,3]

The anomalous Hall (AH) effect measurement has emerged as a powerful tool to gain deep insights into magnetic materials, such as ferromagnetic metals, magnetic semiconductors, and magnetic topological insulators (TIs). In Mn-doped $Bi_2Se_3$, however, the AH effect has never been reported despite a lot of previous studies. Here we report the observation of AH effect in $(Bi,Mn)_2Se_3$ thin films and show that the sign of AH resistances changes from positive to negative as the Mn concentration is increased. The positive and negative AH resistances are found to coexist in a crossover regime. Such a two-component AH effect and the sign reversal can also be obtained by electrical gating of lightly doped samples. Our results provide an important basis for understanding the puzzling interplay between the surface states, the bulk states, and various magnetic doping effects, as well as competing magnetic orders in magnetically doped TIs.

[1] Beijing National Laboratory for Condensed Matter Physics, Institute of Physics, Chinese Academy of Sciences, Beijing 100190, China. [2] School of Physical Sciences, University of Chinese Academy of Sciences, Beijing 100049, China. [3] Beijing Key Laboratory for Nanomaterials and Nanodevices, Beijing 100190, China. Correspondence and requests for materials should be addressed to J.T. (email: jteng@iphy.ac.cn) or to Y.L. (email: yqli@iphy.ac.cn)

Anomalous Hall (AH) effect is one of the most fundamental transport properties of magnetic materials, in which the interplay between magnetism and spin-orbit coupling produces a transverse Hall voltage perpendicular to the applied current and the magnetization[1,2]. Despite more than a century of research, it was not until last decade the AH effect was associated with the Berry curvature of the occupied energy bands[3,4]. Study of the AH effect has since become a valuable tool to understand the electronic properties of magnetic metals and semiconductors, although interpretations of the experimental results are often complicated by the details of band structures as well as the existence of extrinsic sources of AH effect[1,2], namely skew scattering[5] and side jump[6].

The sophisticated nature of AH effect and its connection to the band topology have recently been nicely illustrated in magnetically doped three-dimensional topological insulators (3D TIs, referred to as TIs below)[7,8]. The surface of a TI hosts symmetry-protected gapless Dirac electrons with spin direction locked perpendicular to their momenta. When doped with magnetic impurities, TIs can host a large variety of magnetic phases, such as ferromagnetic, paramagnetic, noncolinear, and spin glass phases, arising from various types of exchange interactions between the magnetic impurities in presence of the surface and bulk states[9–12]. The magnetically doped TIs have received intensive experimental efforts[13], and the work on ferromagnetic (FM) order in Cr-doped $(Bi,Sb)_2Te_3$ thin films has led to the seminal discovery of quantum anomalous Hall effect (QAHE)[14]. In contrast, magnetic doping of $Bi_2Se_3$, a prototype TI isostructural to $Bi_2Te_3$ and $Sb_2Te_3$, has not allowed for the observation of QAHE, even though gapped surface states have been observed in angular resolved photoemission spectroscopy (ARPES) experiments[15,16]. Among them, a particularly puzzling system is the Mn-doped $Bi_2Se_3$, in which even the AH effect has not been discovered despite compelling evidence for FM order in both the bulk and surface states[17–20]. More surprisingly, a recent study combining ARPES and other techniques has strongly suggested a non-magnetic origin for the surface energy gap in the Mn-doped $Bi_2Se_3$, in contradiction to the earlier findings[15,20]. It remains an open question how the non-magnetic

gap, or more generally, the non-magnetic scattering effect of the magnetic impurities competes with magnetic interactions in the Mn-doped $Bi_2Se_3$ and other magnetically doped TIs. Unfortunately, very few experimental techniques can probe the magnetic order in both the surface and bulk states at very low temperatures. Therefore, a systematic study of the AH effect in the Mn-doped $Bi_2Se_3$ thin films will provide much needed information for understanding the complicated impurity-related physics in magnetically doped TIs.

Here, we report a detailed study of the AH effect in the Mn-doped $Bi_2Se_3$ thin films. By controlling the Mn doping level and tuning the chemical potential, the sign of the AH resistance can be varied from positive to negative. The positive and negative AH resistances are found to coexist in a wide range of parameters, and they exhibit qualitatively different dependences on the applied magnetic field and gate voltage. To the best of our knowledge, such a two-component AH effect has never been observed in any magnetically doped TI, despite that both the surface and bulk magnetic orders in TIs have been a subject of intense experimental study. Our results suggest that the non-magnetic scattering effects of the magnetic dopants can have a profound impact on the transport properties of magnetic TIs, and should be taken into account in seeking high quality TI-based magnetic materials.

## Results

**Basic characterization.** The $(Bi_{1-x}Mn_x)_2Se_3$ thin films studied in this work were grown with molecular beam epitaxy (MBE) on $SrTiO_3(111)$ substrates. The samples were then patterned into ~200 μm wide Hall bars (Fig. 1a) for the electron transport measurements. Detailed description of the sample preparation process and the methods for magnetization and transport measurements are given in the Methods section. Figure 1b depicts an atomic force microscopy (AFM) image of a 10 quintuple-layer (QL) thick $(Bi_{1-x}Mn_x)_2Se_3$ film with Mn concentration $x = 0.02$. The surface morphology of the film is characterized by triangular shaped terraces with sizes of the order 100 nm, which are commonly seen in $Bi_2Se_3$ thin films[21]. Figure 1c shows the magnetic field dependence of in-plane

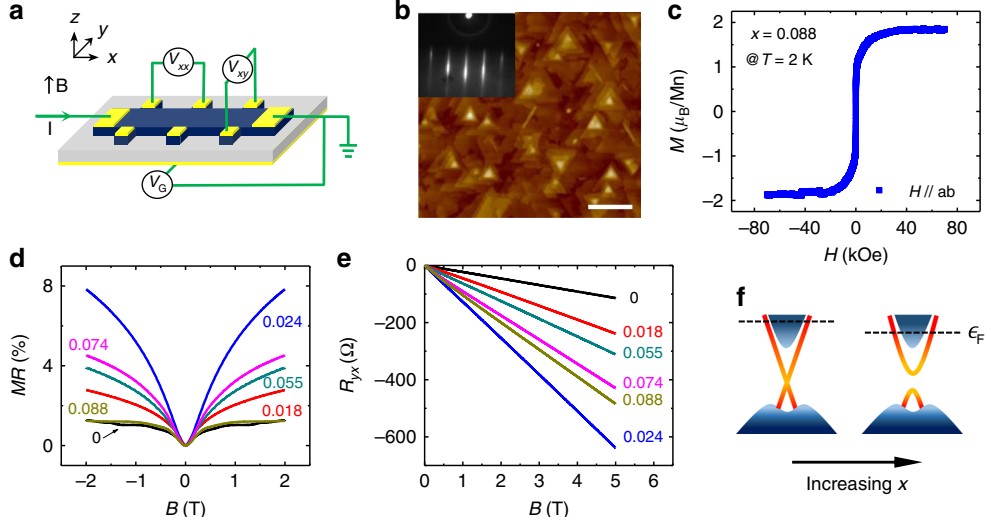

**Fig. 1** Basic characterization of $(Bi_{1-x}Mn_x)_2Se_3$ films. **a** Sketch of a back-gated Hall bar device used in the electron transport measurements. **b** Atomic force microscopy (AFM) and in situ reflection high-energy electron diffraction (RHEED) (inset) images of a 10 quintuple-layer (QL) thick $(Bi_{1-x}Mn_x)_2Se_3$ film ($x$ = 0.02) grown on a STO(111) substrate. **c** In-plane magnetization as a function of applied magnetic field for a 70 nm thick $(Bi_{1-x}Mn_x)_2Se_3$ film with $x$ = 0.088 at $T$ = 1.6 K. The inset shows the temperature dependence of in-plane magnetization. A linear background is subtracted to account for the paramagnetic signal from the STO substrate. Detailed discussion of the magnetic order in the $(Bi,Mn)_2Se_3$ thin films is given in Supplementary Note 3. **d**, **e** Magnetoresistance (MR) and Hall resistance $R_{yx}$ at $x$ = 0, 0.018, 0.0244, 0.055, 0.074, and 0.088 at $T$ = 1.6 K. The MR is defined as $[R_{xx}(B) - R_{xx}(0)]/R_{xx}(0)$. **f** Schematic sketch of band diagrams of the $(Bi,Mn)_2Se_3$ thin films without (left) and with (right) the surface energy gap induced by the Mn doping

magnetization of a 70 nm thick $(Bi_{1-x}Mn_x)_2Se_3$ ($x = 0.088$) thin film. The overall feature of the magnetization is consistent with those reported in refs.[17,20], in which the easy magnetization axes of the $(Bi_{1-x}Mn_x)_2Se_3$ thin films are found to be in the film plane. Figure 1d displays the magnetoresistances (MRs) of a set of $(Bi_{1-x}Mn_x)_2Se_3$ samples with Mn concentration $x = 0$–0.08 (samples A–G). All the samples exhibit positive MR and have a cusp-shaped minimum at $B = 0$ when the gate voltage $V_G$ is set at zero. The corresponding Hall resistance ($R_{yx}$) curves are nearly linear and have negative slopes (Fig. 1e), indicating n-type charge carriers. The sheet electron densities extracted with $n_s = -1/(eR_H)$ are in a range of 0.48–1.29 × 10^13 cm^−2 for the $(Bi_{1-x}Mn_x)_2Se_3$ thin films with $x = 0.018$–0.088 (samples B–G), where $R_H$ is the (ordinary) Hall coefficient. The obtained electron densities are substantially lower than $n_s \sim 2.7 \times 10^{13}$ cm^−2 for an undoped $Bi_2Se_3$ thin film prepared in the similar condition (sample A). The basic transport parameters of these samples are summarized in Table 1, and the corresponding band diagrams are schematically drawn in Fig. 1f. The lower Fermi level in the doped samples can be attributed to the substitution of $Bi^{3+}$ ions by $Mn^{2+}$ ions[19].

**Two-component AH effect.** Figure 2a–f shows the Hall resistance curves for a subset of $(Bi_{1-x}Mn_x)_2Se_3$ thin films ($x = 0$–0.088), and the corresponding AH resistances (Fig. 2g–l) are obtained by subtracting the linear (ordinary) component from

the total Hall resistances, i.e., $R_{AH} \simeq R_{yx} - R_H B$. For the undoped $Bi_2Se_3$ thin film (sample A), the nonlinear part of the Hall resistance is nearly zero for the entire field range. In contrast, the AH resistances are clearly visible for all the Mn-doped films (samples B, D–G). At low doping levels (e.g., $x = 1.8\%$, sample B), the sign of the AH resistance ($R_{AH}$) above the (positive) magnetization saturation field is positive and opposite to that of the ordinary Hall resistance $R_H B$. Throughout this work, we refer to this as the positive $R_{AH}$, which has the same sign as the ordinary Hall resistance of the hole-type carriers. In contrast, the samples with high Mn doping levels ($x = 0.074$ and 0.088) exhibit the negative AH resistance. As shown in Fig. 2, the increase in the Mn concentration drives a crossover from the positive to negative $R_{AH}$. At intermediate doping levels, a kink structure begins to appear in the $R_{AH}$ curves. This suggests coexistence of a positive $R_{AH}$ component with a negative one. Such a two-component AH effect can be observed for a wide range of Mn concentrations ($x \geq 2.4\%$). Figure 2 further shows that these two components have different dependences on the Mn doping level: the magnitude of the negative $R_{AH}$ component becomes larger with increasing Mn concentration, whereas the positive component has a nonmonotonic dependence. The overall trend is that negative component becomes more pronounced relative to the positive component with increasing Mn doping level, as evidenced by the reversal of the sign of $R_{AH}$ in the high magnetic fields.

**Gate-voltage tuning of the AH effect.** The sign reversal in $R_{AH}$ and the two-component AH effect can also be obtained by gate-voltage tuning. This is illustrated with sample C ($x = 0.02$) in Fig. 3. As the gate voltage is decreased from $V_G = +100$ to $-210$ V, the sheet electron density is reduced from $n_s = 0.91 \times 10^{13}$ to $0.35 \times 10^{13}$ cm^−2, based on the Hall effect measurements. The sheet resistance per square (i.e., sheet resistivity $\rho_{xx}$) increases from 2.7 to 6.0 kΩ in the same gate-voltage range. At high electron densities ($V_G \geq 100$ V), $R_{AH}$ only has the positive component. When the electron density is lowered by gating, a kink structure emerges at low magnetic fields and becomes more pronounced. At the lowest electron density, the magnitude of the negative component surpasses that of the positive one. Moreover, the negative $R_{AH}$ component tends to saturate at a magnetic field lower than that of the positive component. Similar phenomena also take place for the

**Table 1 Basic transport characteristics of $(Bi_{1-x}Mn_x)_2Se_3$ ($x = 0.018$–0.088). Here $R_\square$, $n_s$, and $\mu$ denote the sheet resistivity (i.e., resistance per square), sheet electron density, and mobility, respectively. The data were extracted from low field magnetotransport measurements at $T = 1.6$ K**

| Sample ID | Mn fraction (%) | $R_\square$ (kΩ) | $n_s$ (10^13 cm^−2) | $\mu$ (cm²/Vs) |
|---|---|---|---|---|
| A | 0 | 1.19 | 2.74 | 192 |
| B | 1.8 | 1.48 | 1.29 | 327 |
| C | 2.0 | 3.59 | 0.52 | 335 |
| D | 2.4 | 5.27 | 0.48 | 247 |
| E | 5.5 | 2.69 | 1.01 | 230 |
| F | 7.4 | 8.97 | 0.73 | 95 |
| G | 8.8 | 11.8 | 0.65 | 82 |

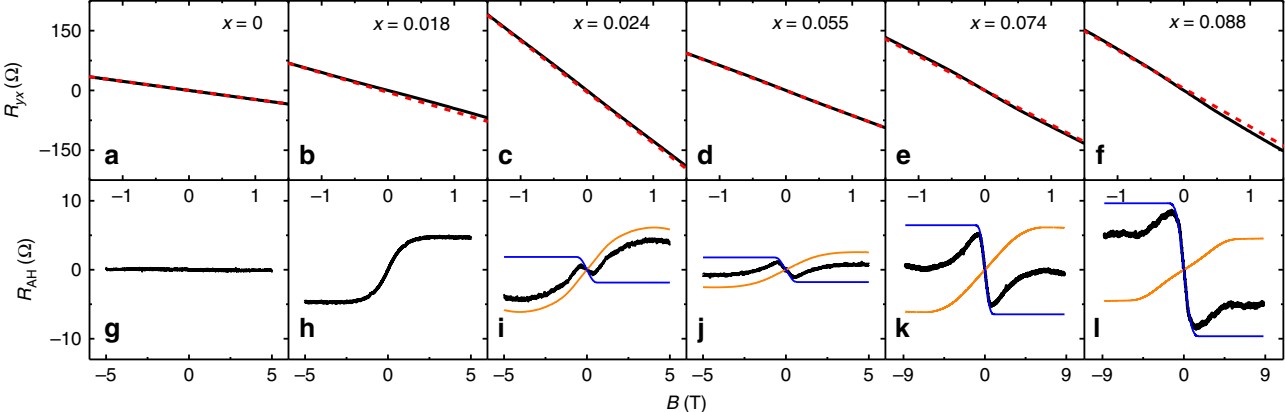

**Fig. 2** Evolution of the Hall effect and the corresponding AH resistances with increasing Mn concentration. Shown in panels **a–f** and **g–l** are the magnetic field dependences of the Hall resistance $R_{yx}$ and the nonlinear part of the Hall resistance ($R_{AH}(B) = R_{yx}(B) - R_H B$), respectively, for Mn concentrations $x = 0$–0.088 at $T = 1.6$ K. For doped samples ($x > 0$), the nonlinear Hall resistances predominantly originate from the AH effect. For the undoped sample ($x = 0$), it is nearly zero. Here the $R_H$ values are obtained by linear fits of the raw $R_{yx}(B)$ curves with magnetic field ranges 3–5 T for samples with $x = 0$–0.055 and 5–9 T for samples with $x = 0.074$ and 0.088. With the methods described in the Supplementary Information, the AH resistance $R_{AH}$ is then separated into a positive component (orange line) and a negative one (blue line) for the samples with $x > 0.02$

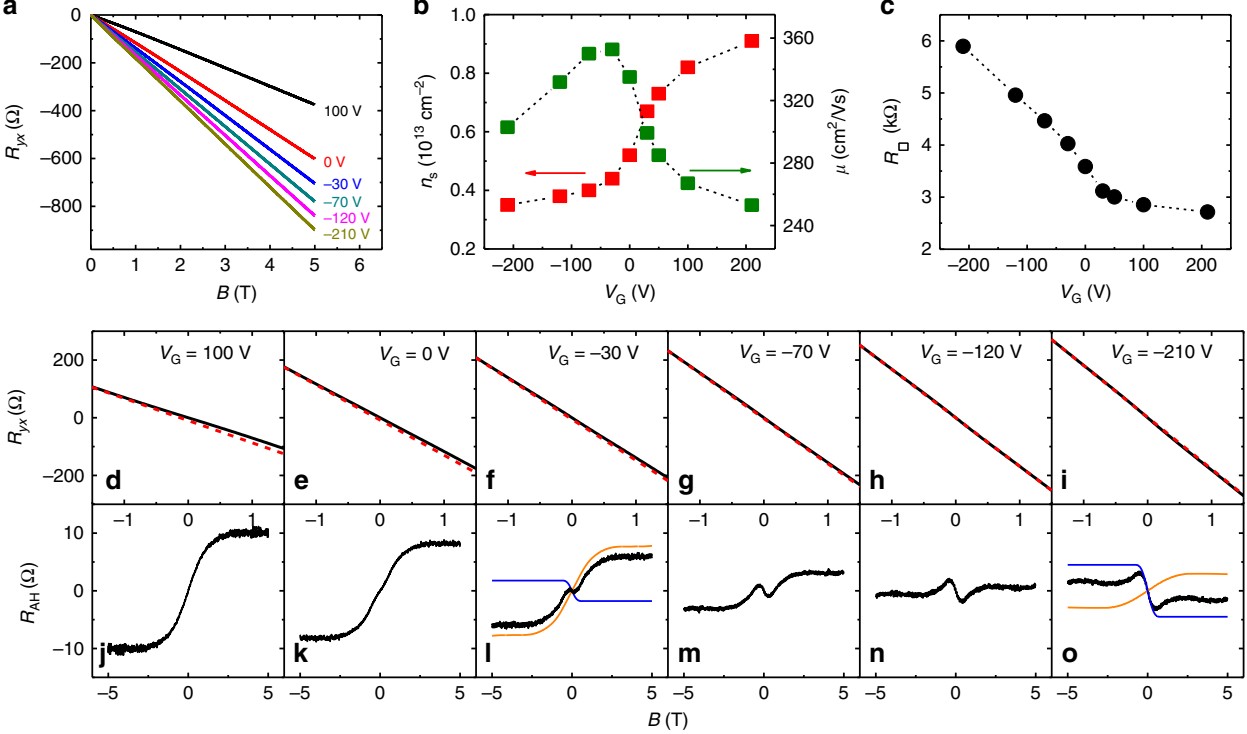

**Fig. 3** Gate-voltage tuning of the anomalous Hall effect in a $(Bi_{1-x}Mn_x)_2Se_3$ thin film ($x = 0.02$). **a** Magnetic field dependences of $R_{yx}$ at different gate voltages. **b** Sheet electron density $n_s$ (red square) and mobility $\mu$ (green square) plotted as a function of gate voltage $V_G$. **c** Gate-voltage dependence of sheet resistivity $R_\square$. **d–i** Magnetic field dependences of $R_{yx}$ at gate voltages from $V_G = 100$ to $-210$ V. **j–o** Magnetic field dependences of $R_{AH}$ for the same range of gate voltages as panels **d–i**. At $V_G = -30$ and $-210$ V, the positive and negative AH components separated from the $R_{AH}$ curves (black) are shown in orange and blue lines, respectively. The Hall effect data were taken from Sample C at $T = 1.6$ K

ungated samples with higher Mn-doping levels ($x = 0.024$–0.088, Fig. 2).

The gate-voltage tuning of the two-component AH effect has also been performed for samples with higher Mn concentrations. Figure 4 shows the results of sample F ($x = 0.074$), in which the sheet electron density can be varied from $n_s = 1.4$ to $0.5 \times 10^{13}$ cm$^{-2}$ and the sheet resistivity from 5 to 65 k$\Omega$. At high electron densities (corresponding to $V_G \geq 100$ V), the magnitude of the positive component is larger than that of the negative component. When the electron density is sufficiently low ($n_s \leq 0.7 \times 10^{13}$ cm$^{-2}$, $V_G \leq 0$ V), the magnitude of negative component increases rapidly with decreasing gate voltage and becomes considerably larger than the positive component. Like the other $(Bi,Mn)_2Se_3$ samples, the negative $R_{AH}$ component in sample F always has a lower saturation field than the positive component, and the relative magnitude between the negative and positive components is significantly enhanced when the electron density is reduced.

**Separation of the two AH components**. The conspicuous differences in the two AH components suggest that they originate from different electronic states. Given that multiple parallel conduction channels coexist in the $(Bi,Mn)_2Se_3$ samples, it is more convenient to analyze the AH effect data in the form of conductivities. In general, the sheet longitudinal conductivity and Hall conductivity are given by $\sigma_{xx} = \sum_i \sigma_{xx,i} = \rho_{xx}/(\rho_{xy}^2 + \rho_{xx}^2)$ and $\sigma_{xy} = \sum_i \sigma_{xy,i} = \rho_{yx}/(\rho_{yx}^2 + \rho_{xx}^2)$, where the summation goes over all conduction channels and $i$ is the channel index. Because of strong impurity scattering effect, most of the transport discussed in this work is in the low mobility regime (i.e., $\mu B \ll 1$), and correspondingly $|\rho_{xy}| \ll \rho_{xx}$. The Hall conductivity can thus

be simplified to $\sigma_{xy} \simeq \rho_{yx}/\rho_{xx}^2$ (with $\rho_{yx} \simeq \rho_{xy}$) and the AH conductivity becomes $\sigma_{AH} \simeq R_{AH}/R_{xx}^2$. Following this definition, $\sigma_{AH}$ has the same sign as the AH resistance $R_{AH}$.

As described in the Supplementary Notes 1 and 2, the AH conductivity can be separated into two components: $\sigma_{AH}(B) = \sigma_{AH,1}(B) + \sigma_{AH,2}(B)$, where the first and second terms denote the positive and negative AH components, respectively. Figure 5 summarizes the results for sample C ($x = 0.02$). As shown in Fig. 5a, the sheet conductivity $\sigma_{xx}$ drops from 9.1 to 4.7 e$^2$/h as $n_s$ is tuned from 0.82 to $0.35 \times 10^{13}$ cm$^{-2}$. Figure 5b,c display the conductivity dependences of $\sigma_{AH,1}^s$ and $\sigma_{AH,2}^s$, the magnitudes of the extracted $\sigma_{AH,1}$ and $\sigma_{AH,2}$ above the saturation fields. The positive component $\sigma_{AH,1}^s$ decreases nearly linearly with decreasing $\sigma_{xx}$, whereas the negative component $\sigma_{AH,2}^s$ become greater until $\sigma_{xx}$ drops to ~6 e$^2$/h. The ratio between the two components, $\sigma_{AH,2}^s/\sigma_{AH,1}^s$, increases monotonically from nearly zero to over 1.6 when $n_s$ is varied from 0.82 to $0.35 \times 10^{13}$ cm$^{-2}$ (Fig. 5d).

Figure 6 summarizes the results for sample F ($x = 0.074$). Figure 6a shows that $\sigma_{xx}$ values of this sample are substantially lower than those of sample C at the same carrier densities. Similar to sample C, $\sigma_{AH,1}^s$ of sample F also has an approximately linear dependence on $\sigma_{xx}$, as depicted in Fig. 6b. However, $\sigma_{AH,2}^s$ of this sample has a drastically different dependence on $\sigma_{xx}$: it becomes smaller as $\sigma_{xx}$ decreases (Fig. 6c). Nonetheless, the $\sigma_{AH,2}^s/\sigma_{AH,1}^s$ ratio of sample F has a qualitatively similar dependence on $\sigma_{xx}$ to sample C, namely increasing monotonically with decreasing $\sigma_{xx}$. As depicted in Fig. 6d, the $\sigma_{AH,2}^s/\sigma_{AH,1}^s$ value increases from about 0.5 to 1.5 as $\sigma_{xx}$ decreases from 5.6 e$^2$/h to about 0.4 e$^2$/h.

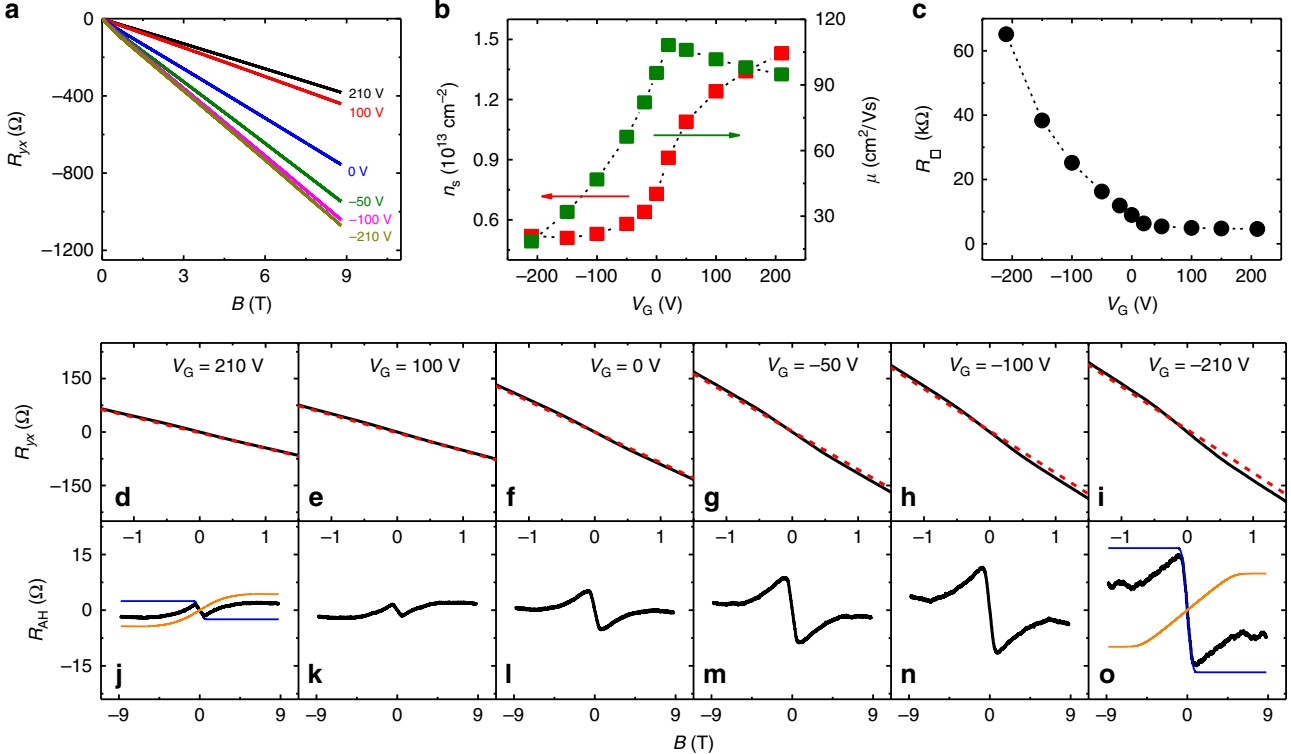

**Fig. 4** Gate-voltage tuning of the anomalous Hall effect in a $(Bi_{1-x}Mn_x)_2Se_3$ thin film ($x = 0.074$). **a** $R_\square$ curves at $V_G = +210$ to $-210$ V. **b** Gate-voltage dependence of $n_s$ (red square) and $\mu$ (green square). **c** Gate-voltage dependence of $R_\square$. **d–i** Magnetic field dependences of $R_{yx}$ for gate voltages from 210 to $-210$ V. **j–o** Magnetic field dependences of $R_{AH}$ for the same range of gate voltages as panels **d–i**. For $V_G = \pm 210$ V, the positive (orange) and negative (blue) AH components separated from the total $R_{AH}$ are also shown. The Hall effect data were taken from Sample F at $T = 1.6$ K

## Discussion

The magnetism in the bulk of Mn-doped $Bi_2Se_3$ thin films have been studied with many techniques, including superconducting quantum interference device (SQUID) magnetometry[17], ferromagnetic resonance[18], polarized neutron reflectivity[18], and X-ray absorption fine structure[22]. These measurements suggest a FM phase in the bulk with a Curie temperature less than 10 K (typically about 5 K) and an in-plane easy axis of magnetization. According to refs. [19,22,23], the Mn impurities are mostly divalent and located in the substitutional Bi sites if the doping level is not too high. It has been proposed that the bulk FM order in (Bi, Mn)$_2$Se$_3$ arise from carrier-mediated exchange coupling of Mn$^{2+}$ ions with 3d$^5$ spin configuration[18], similar to the FM order in the classic diluted magnetic semiconductor (Ga,Mn)As[24].

In contrast to the overall consistent results on the bulk magnetism, the surface magnetism in (Bi,Mn)$_2$Se$_3$ thin films has been controversial. In an early work combining ARPES and X-ray magnetic circular dichroism (XMCD) measurements, Xu et al. suggested a surface FM order with $T_C$ above 100 K and an energy gap up to ~0.1 eV in the surface states due to the magnetic exchange interaction[16]. The gap size is, however, much greater than theoretical values for the Mn-doped TIs[25,26]. The nearly one order of magnitude enhancement of the surface $T_C$ over the bulk value is also at odds with theory that only predicts modest enhancement for the surface magnetism[26]. Recently, Sánchez-Barriga et al. showed with the XMCD measurements that the surface $T_C$ in (Bi, Mn)$_2$Se$_3$ thin films is about 10 K, slightly higher than the FM ordering temperature in the bulk[20]. They also found that in the same films the surface energy gap as large as ~0.2 eV can persist up to room temperature, and more strikingly similar energy gaps were also observed in Bi$_2$Se$_3$ thin films doped with non-magnetic impurities[20]. Based on these findings and previous theoretical work[27,28], it is concluded that the energy gap seen in

(Bi,Mn)$_2$Se$_3$ cannot be attributed to the magnetic exchange interaction, and it rather originates from resonant scatterings between the bulk impurities and the surface states[20]. The ARPES measurements in ref. [20] were, however, carried out at temperatures above $T_C$. It is therefore unclear that how the non-magnetic scatterings competes with the surface FM order in the ground states of (Bi,Mn)$_2$Se$_3$. It is necessary to address both experimentally and theoretically how the non-magnetic interactions introduced by the doping influences the spin structures and electron transport properties in magnetically doped TIs.

Given that the FM order has been confirmed for both the bulk and surface states in (Bi,Mn)$_2$Se$_3$ thin films, it is very puzzling that the AH effect has not been observed in previous transport experiments[17,29]. The AH data presented in this article thus fill an important void in the literature. The distinctively different characteristics of the two AH components strongly suggest they are separately contributed by two sources. On one hand, the positive component $\sigma_{AH,1}$ always gets suppressed when the gate voltage is lowered. Since the (Bi,Mn)$_2$Se$_3$ samples remain n-type for the whole gate-voltage range, decreasing the gate voltage (or making it more negative) always lowers the bulk electron density until the bulk carriers are fully depleted. It is noteworthy that $\sigma_{xx}$ decreases monotonically with decreasing $V_G$. Assignment of $\sigma_{AH,1}$ to the bulk states can thus provide a straightforward explanation of its dependence on $\sigma_{xx}$, since for all known mechanisms of the AH effect in bulk systems, reducing both $\sigma_{xx}$ and $n_s$ would lead to a drop in the AH conductivity[1,2]. It is also very common to observe the AH effect which has the same sign as the ordinary Hall effect in a conventional magnetic system[1,2]. On the other hand, for the samples with low Mn concentrations (e.g., $x = 0.02$), the negative component $\sigma_{AH,2}$ becomes larger with decreasing gate voltage. As will be detailed below, the sign and the chemical potential dependence of $\sigma_{AH,2}$ qualitatively agree with a previous

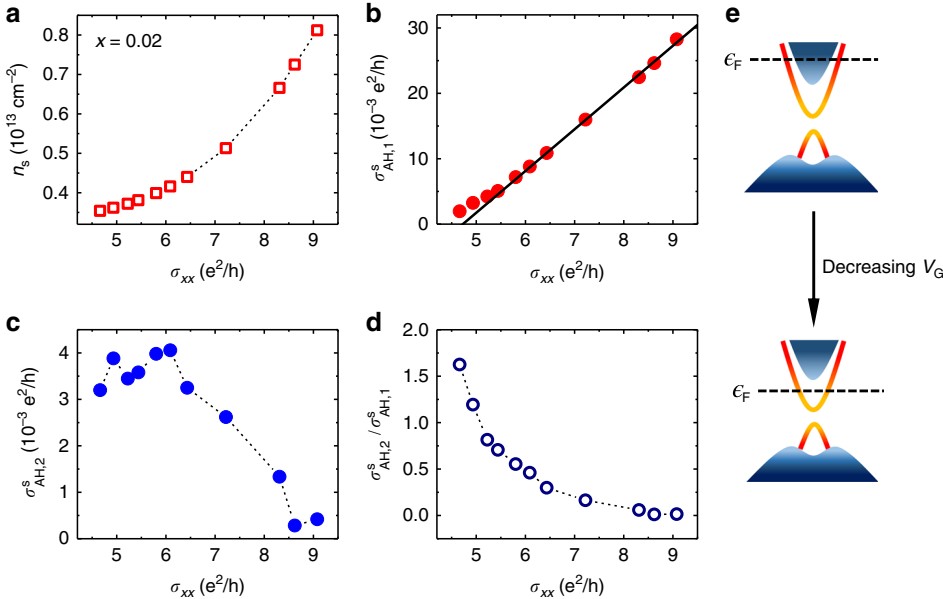

**Fig. 5** Characteristics of the AH conductivity in a lightly doped $(Bi_{1-x}Mn_x)_2Se_3$ sample ($x = 0.02$). **a** Sheet carrier density $n_s$ as a function of sheet conductivity $\sigma_{xx}$. **b,c** $\sigma_{xx}$ dependences of the magnitudes of the positive and negative AH conductivities above the saturation fields, $\sigma_{AH,1}$ (panel **b**) and $\sigma_{AH,2}$ (panel **c**). **d** The ratio of the AH components, $\sigma_{AH,2}^s/\sigma_{AH,1}^s$, plotted as a function of $\sigma_{xx}$. **e** Schematic band diagrams for high and low Fermi levels. The data shown in panels **a**–**d** were extracted from the Hall effect data taken from Sample C at $T = 1.6$ K

experiment on Mn-doped $Bi_2(Te,Se)_3$ thin films, in which the AH effect is dominated by the intrinsic contribution of the surface states, at least at low carrier densities[9]. Assignment of $\sigma_{AH,2}$ to the surface states is also consistent with the prediction that perpendicular anisotropy is favored for the magnetization of the TIs surface states[26,30,31]. As described in the previous section, $\sigma_{AH,2}$ has a saturation field much lower than the positive component $\sigma_{AH,1}$, which arises from the bulk FM order with the in-plane anisotropy (Please see Supplementary Note 3 and Supplementary Figures 9 and 10 for further discussion of the magnetic order in $(Bi,Mn)_2Se_3$ thin films).

The chemical-potential dependence of AH conductivity in the TI surface states has been explained with the theories based on the massive Dirac fermion model[32–35]. On the simplest level, the ferromagnetically ordered magnetic impurities are treated as an exchange energy term $\Delta = J\mathbf{S} \cdot \mathbf{s}$ in the Hamiltonian of surface states, where $J$ is the exchange constant between the impurity spin $\mathbf{S}$ and surface electron spin $\mathbf{s}$. This term turns the gapless surface states into a gapped 2D Dirac band with $\epsilon = \pm\left[(\hbar ck)^2 + \Delta^2\right]^{\frac{1}{2}}$, where $c$ is the Fermi velocity and $k$ is the Fermi wavevector. The energy gap is thus $2\Delta$, corresponding to an effective mass $m = \frac{\Delta}{c^2}$, which could be positive or negative, depending on the sign of exchange constant J. In the mean field level, the intrinsic AH conductivity is given by $\sigma_{AH}^{int} = -\frac{e^2}{2h}\frac{\Delta}{\epsilon_F}$, where $\epsilon_F$ is Fermi energy. Contributions from extrinsic sources, i.e., side jump and skew scatterings, can modify the AH conductivity substantially. According to Ado et al.[35], the total AH conductivity in the weakly disorder limit is given by

$$\sigma_{AH}^{tot} = -\frac{8e^2}{h}\frac{\epsilon_F\Delta^3}{(\epsilon_F^2 + 3\Delta^2)^2} \quad (1)$$

The magnitude of $\sigma_{AH}^{tot}$ decreases monotonically from $\frac{1}{2}\frac{e^2}{h}$ toward zero as $\epsilon_F$ increases from $|\Delta|$ (corresponding to the upper gap edge) to larger values. In the limit of $\epsilon_F \gg \Delta$, Eq. (1) reduces to $\sigma_{AH}^{tot} \propto \left(\frac{\Delta}{\epsilon_F}\right)^3$. In the $(Bi,Sb)_2(Te,Se)_3$ family of magnetic TIs,

the sign of $m$ (and hence the sign of intrinsic surface AH effect) can be tuned by varying the chemical composition. For instance, the $(Bi,Sb)_2Te_3$ thin films doped with Cr[14], V[36], or Mn[37] exhibit positive AH resistances, whereas the Mn-doped $(Bi,Sb)_2(Te,Se)_3$ has an opposite sign[9]. These results suggest that the sign difference may be attributed to the details of the p–d exchange interactions between the Mn ions and the anions ($Te^{2-}$ or $Se^{2-}$). For the samples with low Mn concentration, the gate-voltage dependence of the negative component $\sigma_{AH,2}$ is qualitatively in agreement with the above theory. As shown in Fig. 5, when $\epsilon_F$ in sample C ($x = 0.02$) is lowered by gating, $\sigma_{AH,2}^s$ increases by an order of magnitude. Based on the Hall effect data, as well as the ARPES results reported in ref. [20], $\epsilon_F \approx 0.3$ eV can be estimated for the large positive gate voltages. Following Eq. (1) and using the $\sigma_{AH,2}^s$ values in Fig. 5, one obtains a magnetic gap $\Delta$ of the order 10 meV, which is comparable to the non-magnetic energy gap $\Delta_{nm}$ observed in $(Bi_{1.98}Mn_{0.02})_2Se_3$ thin film with ARPES measurements[20]. The fair agreement between the experiment and the massive Dirac fermion model can be attributed to $\epsilon_F \gg \Delta_{nm}$, which makes the non-magnetic effect of impurities insignificant.

For the samples with high Mn concentration or with very low Fermi levels, the surface AH conductivity can no longer be described with the massive Dirac fermion model. As shown in Fig. 6, $\sigma_{AH,2}^s$ in sample F ($x = 0.074$) becomes smaller with decreasing electron density, opposite to the chemical potential dependence given by Eq. (1). According to ref. [20], a Mn concentration of $x = 0.074$ would lead to a non-magnetic gap of $2\Delta_{nm} \approx 0.2$ eV, which is comparable to the Fermi energy $\epsilon_F$ for the entire range of gate voltages. For sample C ($x = 0.02$), the observed saturation of $\sigma_{AH,2}^s$ at large negative gate voltages can also be attributed to the chemical potential being comparable to $\Delta_{nm}$. The strong resonance scatterings between the magnetic impurities in the bulk and the surface states have been identified to the origin of the robust non-magnetic gap observed in the ARPES experiment[20]. The ground state spin structure is expected to deviate significantly from the standard hedgehog spin texture derived from the massive Dirac fermion model[32], because of such

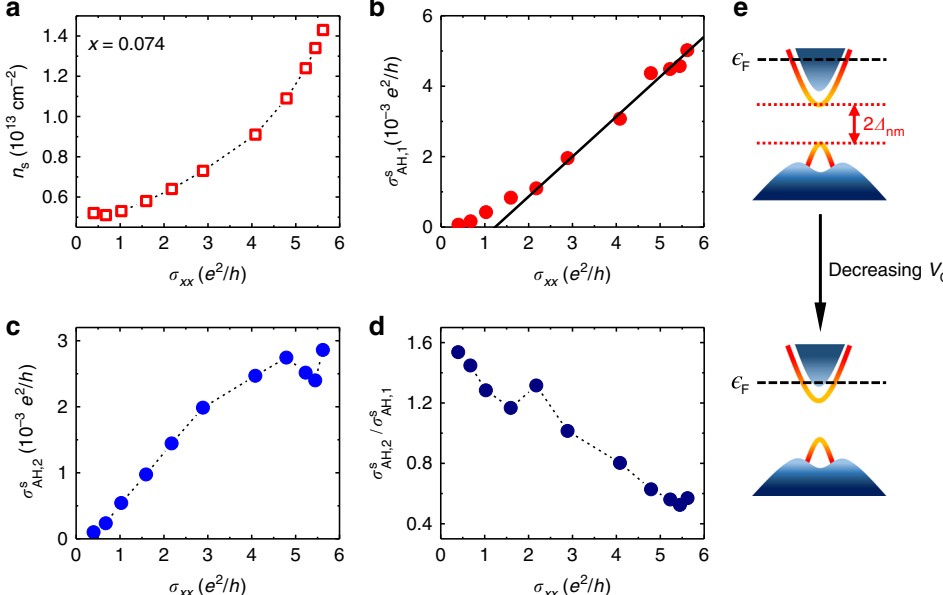

**Fig. 6** Characteristics of the AH conductivity in a heavily doped $(Bi_{1-x}Mn_x)_2Se_3$ sample ($x = 0.074$). Plotted in panels **a–d** are $\sigma_{xx}$ dependences of $n_s$, $\sigma^s_{AH,1}$, $\sigma^s_{AH,2}$, and the ratio $\sigma^s_{AH,2}/\sigma^s_{AH,1}$, respectively. **e** Schematic band diagrams of the (Bi,Mn)$_2$Se$_3$ sample with high and low Fermi levels. The data shown in panels **a–d** were extracted from the Hall effect data taken from Sample F at $T = 1.6$ K

impurity scattering effects of the bulk dopants. This can account for the suppression of the surface state AH conductivity ($\sigma^s_{AH,2}$) shown in Figs. 5 and 6 (as well as in Supplementary Figure 8), when the $\epsilon_F$ becomes comparable to $\Delta_{nm}$. It is noteworthy that the non-magnetic impurity scatterings have also been invoked to explain the absence of the energy gap on the TI surfaces covered by magnetic dopants[27]. According to recent theoretical studies, such processes can also cause the gap filling effect[27], and even the sign reversal of the AH conductivity in magnetically doped TIs[38]. In short, magnetic doping can lead to many intriguing quantum phenomena that requires further investigation into the microscopic connection between the surface and bulk states in TIs.

Finally, it is worth noting that the sign reversal of AH conductivity has also been observed in other magnetic systems, such as field effect structures of (Ga,Mn)As thin films[39] and heavily doped TI thin films $Bi_{1.78}Cr_{0.22}(Te_{1-y}Se_y)_3$[40]. In these systems, however, the AH conductivity has only one component, either positive or negative, for a certain gate voltage and/or chemical composition. The sign reversal of the AH effect in the (Bi, Mn)$_2$Se$_3$ thin film is caused by the gate-voltage or doping induced change in the relative weight of the two AH components, which rather originate from different sources, namely the bulk states and surface states. The observed chemical potential dependence of the negative AH component reveals, for the first time, an important role of the non-magnetic potential scatterings of the magnetic impurities in the transport properties of the surface states in the magnetically doped TIs. It will also be very interesting to explore whether the interplay between the drastically different surface and bulk magnetizations, along with the competition from various impurity effects, could lead to novel spin structures, such as spin canting, noncolinear, or topological spin textures[11,41].

## Methods

**Thin film growth and characterization**. The $(Bi_{1-x}Mn_x)_2Se_3$ thin films were grown on SrTiO$_3$ (111) substrates in a molecular beam epitaxy (MBE) system with base pressure of $3 \times 10^{-10}$ mbar. For the growth of high quality thin films, it is important to have atomically clean and flat substrates. The STO substrates were cleaned with solvents and deionized water, followed by annealing at 1150 °C for 4 h in high pure oxygen (99.999%) before being transferred into the MBE chamber. During the growth, the substrate temperature was kept at 200 °C with a tungsten-

filament heater mounted on the back of the sample holder. High-purity Bi, Mn, and Se sources (99.999%) were used and the evaporation rates were controlled by adjusting the temperatures of effusion cells. The flux ratio of Bi and Se was set at about 1:20, and the magnetic doping level was regulated by the flux ratio between Mn and Bi fluxes. In this work, the Mn concentrations in the $(Bi_{1-x}Mn_x)_2Se_3$ thin films were limited to $x \leq 8.8\%$. According to ref. [19], such low Mn doping levels are necessary for obtaining samples free from secondary phases. The growth process was monitored with an in situ reflection high-energy electron diffraction (RHEED), and the high quality of the epitaxial growth was confirmed by streaky RHEED patterns. The thicknesses of the (Bi,Mn)$_2$Se$_3$ films were chosen to be 10 nm to obtain good gate-tunability while avoiding the hybridization effect of the top and bottom surfaces. The $(Bi_{1-x}Mn_x)_2Se_3$ samples were characterized by AFM, X-ray diffraction, and vibrating sample magnetometry.

**Transport measurements**. After being taken out of the MBE chamber, the $(Bi_{1-x}Mn_x)_2Se_3$ samples used for electron transport measurements were quickly patterned into Hall bars by hand scratching in order to minimize the surface degrading effects. Electrical contacts as well as the backgate electrodes were prepared with thermally deposited of Cr/Au bilayers. The transport measurements were carried out in a helium vapor flow cryostat with temperatures down to 1.6 K and magnetic fields up to 9 T with standard lock-in technique. The samples were typically excited by a 100 nA ac current with a frequency of ~13.7 Hz. The backgate voltage was applied with a source-measurement unit with the leakage current limited below 2 nA.

**Analyses of the Hall effect data**. The AH resistances are obtained by subtracting the ordinary Hall resistances from the raw Hall effect data. The method is described in detail in Supplementary Note 1 and illustrated in Supplementary Figures 1 and 2. Its validity is further confirmed with the temperature dependent data shown in Supplementary Figures 2 and 3. Additional AH data extracted for two lightly doped samples (Samples D and H) are depicted in Supplementary Figure 4. Two methods for separating the two AH components are described in Supplementary Note 2 and illustrated in Supplementary Figures 5 and 6. The consistency between these two methods is shown in Supplementary Figures 7 and 8.

**Data availability**. The data that support the findings of this study are available from the corresponding author upon request.

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

## Acknowledgements

We thank H. Z. Lu, S. Q. Shen, and P. Ostrovsky for valuable discussions. This work was supported by the National Key Research and Development Program (Project No. 2016YFA0300600), the National Science Foundation of China (Projects No. 61425015, No. 11604374, and No. 11374337), the National Basic Research Program of China (Projects No. 2015CB921102), and the Strategic Initiative Program of Chinese Academy of Sciences (Projects No. XDPB0801, No. XDB070200, and No. XDPB0602).

## Author contributions

Y.L. and J.T. initiated the project. L.N. and J.T. carried out the MBE growth. L.N. fabricated the devices and performed the electron transport measurements. L.N. and Y.L. analysed the data. All of the authors prepared the manuscript.

## Additional information

**Competing interests:** The authors declare no competing interests.

