## [Peer Review File(PDF 855 kb) · Nature Communications]

Reviewers' comments:

Reviewer #1 (Remarks to the Author):

This work investigates $(\text{Bi}_{1-x}\text{Mn}_x)_2\text{Se}_3$ which is a prototype magnetically doped topological insulator. This material has previously been investigated in many studies in the literature, nevertheless the anomalous Hall effect (AHE) has never been observed in this material, in contrast to related magnetic topological insulators.

In the present work the authors grow epitaxial films of $(\text{Bi}_{1-x}\text{Mn}_x)_2\text{Se}_3$ on SrTiO_3 which has the additional advantage that they can perform magnetotransport not only as a function of the Mn composition of the film but also of a gating voltage.

As a result they report that they observe the outstanding AHE and that it changes sign with the Mn concentration. In addition, they also report a gating voltage dependence of this sign.

They argue that the sign change is due to a shift of the chemical potential from the bulk gap region (but outside of the nonmagnetic gap of the surface state) to the bulk conduction band.

In my view, this manuscript leaves more questions open than it answers.

First of all, the AHE is a property that has to be extracted after the Hall effect is subtracted. The authors describe the procedure but it is still not clear to me how much this is tolerant to faults.

Secondly, $(\text{Bi}_{1-x}\text{Mn}_x)_2\text{Se}_3$ has in-plane magnetic anisotropy and I think this is the reason why the AHE has never been observed in this system. But also in the present work I do not see any indication how the out-of-plane anisotropy has been achieved. The magnetization data in Fig. 1c show in-plane magnetization.

The shift of the chemical potential is schematically shown and it is an important part of the argumentation. Note that the sign change of the carriers with Mn concentration in $(\text{Bi}_{1-x}\text{Mn}_x)_2\text{Se}_3$ has only been reported for bulk single crystals (Choi et al. Ref. 29) and not for MBE grown films.

At least for the Mn concentration dependence (maybe not for the gating dependence) the shift of the chemical potential could be demonstrated by ARPES. It requires probably sending the samples to an ARPES group but for a Nature Communications publication I expect some effort to obtain a complete picture. Especially for a rather disputed system as the present $(\text{Bi}_{1-x}\text{Mn}_x)_2\text{Se}_3$.

Therefore I do not recommend publication in Nature Communications.

Reviewer #2 (Remarks to the Author):

This paper reports on an interesting transport and magnetic property study of Bi_2Se_3 thin films that are doped with Mn at different doping levels. The main conclusion is that the sign of the anomalous Hall effect changes with Mn concentration, and this is interpreted in terms of competing surface and bulk contributions.

As far as I can tell there is no evidence of magnetic order in these films, and perhaps not should be expected, in spite of the discussion in the MS. The magnetization measurements seem to show that

Mn supplies weakly interacting local moments that are aligned by an external magnetic field.

The most interesting aspect of this experiment is its study of transport properties.

In my opinion there is an opportunity to improve the discussion of how these measurements, the Hall measurements, are interpreted. I recommend against publication of the MS in its present form. The main problem is that a lot is made out of small deviations from linearity in the Hall effect vs. magnetic field data, and the dependence of

those deviations on Mn concentration. The conclusions are quite sensitive to how the data is analyzed and this is not explained in sufficient detail. Indeed there is a risk that the observations may be over interpreted since

the deviations from linearity are at the 5% level. I fear that there may be small deviations from linearity whose origin is different from what the authors assume.

Here is how I would interpret the data - (and indeed I think that this is exactly what the authors have in mind but it is not spelled out explicitly nor implemented with an adequate degree of self-criticism.)

In the absence of Mn σ_H is linear in field - this is presumably the ordinary Hall effect. Once the magnetic field is strong enough to

align the Mn spins there are two contributions - one from the Hall effect and one from the ordinary Hall effect. Importantly it is easily possible that the ordinary Hall effect - ratio of σ_H to B - with saturated Mn spins could change slightly from its value when the

Mn spin orientations are random. The sign and size of the AHE is identified ultimately from the offset between the linear behaviors at positive and negative magnetic fields strong enough to saturate the magnetization in different directions. The shape of the

AHE vs. field curves don't make much sense at intermediate Mn concentrations, and this is I believe a hint at the uncertainty of the interpretation.

I recommend that the authors be more forthright about the uncertainties in the interpretation, and explain in more detail, the choices they have made in separating the Hall conductivity into ordinary and anomalous Hall contributions.

The data are interesting and extensive, and do address an important puzzle. The introduction contains a useful summary of relevant literature. I recommend that a MS with a carefully reconsidered analysis which points readers to all the danger points be accepted for publication.

Reviewer #3 (Remarks to the Author):

The manuscript titled by "Two-component anomalous Hall effect in a magnetically doped topological insulator", reports the observation of abnormal anomalous Hall (AH) effect in Mn doped Bi₂Se₃ films. The authors have carried out detail investigation of magneto-transport measurement in a series of 10 quintuple-layer (QL) thick (Bi_{1-x}Mn_x)₂Se₃ film with Mn concentration ranging from x = 0 to x = 0.088. The Hall effect is studied as the function of Mn concentration, and of carrier concentration (Fermi level) which is controlled by gating. The paper reports that the sign of AH resistances can be changed from positive to negative by the Mn concentration, and these two types of AH resistances are found to coexist in the crossover regime. Such a two-component AH effect and the sign reversal can also be obtained by lowering the chemical potential in the samples with low Mn-doping levels. Based on their different dependences on the gate voltage and magnetic field, the authors assigned the positive and negative AH components to the bulk and surface states, respectively. The separation of the two AH components in the resistance data is introduced and the mechanism of these observation are also

discussed. The topic of paper is very interesting and the data are impressive. However, I cannot recommend the publication of the paper as its current form because of two major issues.

1. There is no temperature dependent AH resistance data and magnetization data presented in the paper. Thus, it is hard to conclude the observed curvature in the Hall data is due to the magnetization in the films. Clearly, it is well known that there are two kinds of carriers (bulk and surface) in the Bi₂Se₃ films, and these carriers also have different mobility. The Mn concentration and gating can tune the ratio of these two kinds of carriers; therefore, applying a single carrier model (linear Hall resistance) for normal Hall resistance is unjustified.

2. Does the observed magnetization curves have hysteresis loop? Does AH resistance data have hysteresis loop? Are two of them comparable? Although the paper is focusing on high field data, the authors should establish the basic properties of AH resistance as the starting point.

In addition to the justification of AH data, there are some minor technical issues the authors should also consider when they revised the data.

1. The methods of the separation of two AH components are illustrated in Figs 7 and 8. The second method of the separation is not well described in Figure. Fig. 8 only serves as results for validity. Could the authors produce similar illustration as Fig. 7.

2. The mobility data, both as function of Mn concentration and gating voltage, should be provided.

3. The mechanism of opposite sign for bulk and surface AH resistance is not well explained.

4. The different behavior of surface AH conductivity as function of sheet conductivity (Fig. 5c and Fig. 6c) is very interesting. Is this a general rule, bulk AH conductivity is monotonically increasing as sheet conductivity (Fig 5b and Fig 6b)? On the other hand, surface AH conductivity has a highest point around $5-6 e^2/h$, hasn't it?

5. Have the authors carried out (field orientation) angular dependent of magneto-transport measurement to separate the conductivity of surface and bulk effect?

Response to the reviewers' comments

We thank all reviewers for their time and efforts. We have carefully considered their comments, which are valuable in improving the quality of our manuscript. Please see below for the responses to each of the individual comments, as well as a list of changes in the manuscript.

Reviewer #1

Comments: This work investigates $(\text{Bi}_{1-x}\text{Mn}_x)_2\text{Se}_3$ which is a prototype magnetically doped topological insulator. This material has previously been investigated in many studies in the literature, nevertheless the anomalous Hall effect (AHE) has never been observed in this material, in contrast to related magnetic topological insulators. In the present work the authors grow epitaxial films of $(\text{Bi}_{1-x}\text{Mn}_x)_2\text{Se}_3$ on SrTiO_3 which has the additional advantage that they can perform magnetotransport not only as a function of the Mn composition of the film but also of a gating voltage. As a result they report that they observe the outstanding AHE and that it changes sign with the Mn concentration. In addition, they also report a gating voltage dependence of this sign. They argue that the sign change is due to a shift of the chemical potential from the bulk gap region (but outside of the nonmagnetic gap of the surface state) to the bulk conduction band. In my view, this manuscript leaves more questions open than it answers.

Reply: We thank the reviewer for giving a nice summary of the main results of our experiment. The most important aspect of our manuscript is to report the first observation of the anomalous Hall effect (AHE) in Mn-doped Bi_2Se_3 and to provide the first evidence for a two-component AHE in any magnetic TIs (and perhaps in any magnetic material, to the best of our knowledge). This is well supported by our experimental data, which were unfortunately not presented with sufficient technical details in the previous manuscript. This might cause the reviewer having an impression that "it leaves more questions open than it answers". In response to the criticisms of the reviewer, we have modified the manuscript and provided an 18-page Supplementary Information, which contains a lot of additional data and technical notes. In the following, we address each specific comment of the reviewer.

First of all, the AHE is a property that has to be extracted after the Hall effect is subtracted. The authors describe the procedure, but it is still not clear to me how much this is tolerant to faults.

Reply: In addition to the control experiment that shows the absence of the AHE at $T = 1.6$ K in an undoped Bi_2Se_3 thin film grown under the same condition (Fig. 2a), we have also carried out transport measurements of both doped and undoped samples at various temperatures to rule out other possibilities. As shown in Fig. S2, the nonlinear part of the Hall resistance has a strong temperature dependence for a Mn-doped Bi_2Se_3 thin film, whereas it is nearly independent of temperature for the undoped Bi_2Se_3 thin film. The AH resistance, obtained by subtracting the nonlinear component at $T = 150$ K, has a temperature dependence and a magnetic field dependence consistent with what is expected for the magnetization of a thin film with an in-plane easy magnetization axis and low magnetic ordering temperature. The extracted AH component diminishes at $T > 25$ K, at which the $(\text{Bi,Mn})_2\text{Se}_3$ thin film becomes paramagnetic. Moreover, the sharp kink structure observed at some Mn concentrations or gate voltages (see Figs. 2 & 3) can be unequivocally attributed to a magnetic origin since such features have never been seen in a non-magnetic TI thin film (See Fig. S1 and Supplementary Note 1). In short, our systematic measurements of the Hall effect at various temperatures, Mn concentrations (including zero Mn-doping), and gate voltages can coherently establish the conclusion that the nonlinear Hall effect components are contributed significantly the Mn-doping, at least for the magnetic fields relevant to this work. The nonlinear Hall resistances due to the AHE can be differentiated from those from other sources, such as the coexistence of multiple types of carriers. Please see Supplementary Note 1 and Figs. S1-S10 for more details.

Secondly, $(\text{Bi}_{1-x}\text{Mn}_x)_2\text{Se}_3$ has in in-plane magnetic anisotropy and I think this is the reason why the AHE has never been observed in this system. But also in the present work I do not see any indication how the out-of-plane anisotropy has been achieved. The magnetization data in Fig. 1c show in-plane magnetization.

Reply: The bulk of $(\text{Bi}_{1-x}\text{Mn}_x)_2\text{Se}_3$ has an in-plane magnetic anisotropy, which indeed makes it not easy to observe the corresponding AHE. In our experiment, however, we performed the transport measurements in the magnetic fields high enough to overcome the anisotropy field, so that the magnetization acquires a significant perpendicular component (due to the magnetization rotation and other dynamics). At the same time, the ordinary Hall effect still maintains a good linear dependence on the magnetic field due to the low carrier mobilities (See also Figs. S1 & S2, and Supplementary Note 1). Such a careful balance of experimental parameters enabled us to observe the AHE from the bulk of $(\text{Bi,Mn})_2\text{Se}_3$ thin films for the first time. In addition, our measurements of the samples with different Mn-concentrations and gate voltages allowed for the observation of a negative AHE component, which has a much lower saturation field than that of the positive counterpart. The chemical potential and conductivity dependences of the negative component strongly suggest

the negative component originates from the surface states. As revealed in previous theoretical and experimental studies (Refs. [9,26,31]), the surface ferromagnetism in TIs tends to have an out-plane anisotropy in magnetic TIs. This is consistent with the much lower saturation field for the negative AH component. Therefore, our experiment strongly suggests the coexistence of in-plane and out-of-plane anisotropies in the Mn-doped Bi_2Se_3 thin films, even though we do not rule out a competition between them due the small thickness of the films (Please see Supplementary Note 3 for more discussion). This is quite unique among all the magnetically doped TIs studied to date.

The shift of the chemical potential is schematically shown and it is an important part of the argumentation. Note that the sign change of the carriers with Mn concentration in $(\text{Bi}_{1-x}\text{Mn}_x)_2\text{Se}_3$ has only been reported for bulk single crystals (Choi et al. Ref. 29) and not for MBE grown films. At least for the Mn concentration dependence (maybe not for the gating dependence) the shift of the chemical potential could be demonstrated by ARPES. It requires probably sending the samples to an ARPES group but for a Nature Communications publication I expect some effort to obtain a complete picture. Especially for a rather disputed system as the present $(\text{Bi}_{1-x}\text{Mn}_x)_2\text{Se}_3$. Therefore I do not recommend publication in Nature Communications.

Reply: A well-executed ARPES measurement can indeed provide valuable information on the chemical potential in a Mn-doped Bi_2Se_3 thin film. We would have collaborated with one of the ARPES groups in our institute if such measurements had not been performed very nicely by the group of Rader in Berlin. As shown in Ref. [20], the Fermi level shifts about 0.1 V downward as the Mn-doping level is increased from $x = 0$ to 0.08. This is fully consistent with the overall trend in the chemical potentials estimated from our Hall effect measurements of the $(\text{Bi},\text{Mn})_2\text{Se}_3$ thin films (Fig. 2 and Table 1). It is noteworthy that an *ex situ* ARPES measurement of our films would require protecting them with selenium capping in the MBE system followed by a decapping process in the ARPES system. However, it was pointed out in Ref. [20] that such a decapping process could modify the properties of the $(\text{Bi},\text{Mn})_2\text{Se}_3$ thin films and lead to spurious signals. Therefore, we decided not to repeat the ARPES work reported in Ref. [20], and rather use their data to assist the analysis of our transport results.

Reviewer #2

Comments: This paper reports on an interesting transport and magnetic property study of Bi_2Se_3 thin films that are doped with Mn at different doping levels. The main conclusion is that the sign of the anomalous Hall effect changes with Mn concentration,

and this is interpreted in terms of competing surface and bulk contributions.

Reply: We thank the reviewer for finding our work interesting and giving a lot of valuable critical remarks, which have helped us to improve our manuscript significantly. Please see below for the responses to each of the technical comments.

As far as I can tell there is no evidence of magnetic order in these films, and perhaps not should be expected, in spite of the discussion in the MS. The magnetization measurements seem to show that Mn supplies weakly interacting local moments that are aligned by an external magnetic field.

Reply: The magnetic properties of $(\text{Bi,Mn})_2\text{Se}_3$ thin films have been studied previously by many other groups. Evidence for ferromagnetic order in both bulk and surface states has been reported [Refs. 17-20]. Our magnetization measurement with SQUID magnetometer (Fig. 2c), which is only sensitive to the bulk signal, is fully consistent with the previous experiments on the $(\text{Bi,Mn})_2\text{Se}_3$ films grown with MBE. Despite that the magnetization curve (M vs H) has no obvious hysteresis and its shape somewhat resembles that of a paramagnetic or superparamagnetic system, detailed measurements (see for instance the Arrott plot (Fig. 3c) in Ref. [20]) showed that the ground state is ferromagnetic and Curie temperature is about several Kelvin. This agrees with our AHE measurements. As shown in Fig. S2, the low field slope of anomalous Hall resistance, dR_{AH}/dB , deviates from the linear dependence on $1/T$ at about 5 K, supporting the magnetic ordering at low temperatures. For a paramagnetic system, one would expect the linear law for all temperatures. Additional evidence for ferromagnetic order in the bulk has been obtained with ferromagnetic magnetic resonance (FMR) measurements [see Refs. 18]. Furthermore, the X-ray magnetic circular (XMCD) measurements reported in Ref. [20] suggest surface ferromagnetic order with a T_C of about 10 K. This is also consistent with the temperature dependence of the negative AH component (See Supplementary Fig. 3), which is related to the surface states. The $(\text{Bi,Mn})_2\text{Se}_3$ films used in our work are similar to those in Ref. [20], it is thus not necessary to repeat most of the previous characterizations of magnetic properties. This enabled us to focus better on the AHE in Mn-doped Bi_2Se_3 thin films, which has never been observed and appears to be quite unique due to the coexistence of two components with opposite signs.

The most interesting aspect of this experiment is its study of transport properties. In my opinion there is an opportunity to improve the discussion of how these measurements, the Hall measurements, are interpreted. I recommend against publication of the MS in its present form. The main problem is that a lot is made out of small deviations from linearity in the Hall effect vs. magnetic field data, and the dependence of those deviations on Mn concentration. The conclusions are quite

sensitive to how the data is analyzed and this is not explained in sufficient detail.

Reply: We agree with the reviewer that our interpretation of the experimental data needs to be improved. We have composed an 18-page Supplementary Information to describe in detail how the AH resistances can be extracted reliably, and spurious effects can be ruled out with systematic control experiments. Please see Supplementary Figs. S1-S10 and Notes 1-3 for details.

Indeed there is a risk that the observations may be over interpreted since the deviations from linearity are at the 5% level. I fear that there may be small deviations from linearity whose origin is different from what the authors assume. Here is how I would interpret the data - (and indeed I think that this is exactly what the authors have in mind but it is not spelled out explicitly nor implemented with an adequate degree of self-criticism.) In the absence of Mn, σ_H is linear in field - this is presumably the ordinary Hall effect. Once the magnetic field is strong enough to align the Mn spins there are two contributions - one from the Hall effect and one from the ordinary Hall effect. Importantly it is easily possible that the ordinary Hall effect - ration of σ_H to B - with saturated Mn spins could change slightly from its value when the Mn spin orientations are random. The sign and size of the AHE is identified ultimately from the offset between the linear behaviors at positive and negative magnetic fields strong enough to saturate the magnetization in different directions.

Reply: We thank the referee for providing some deep thoughts on our data. The suggested alternative scenario for the nonlinear part of the Hall effect can, however, be excluded with our measurements in tilted magnetic fields. The alignment of localized spins by the magnetic field can in principle modify carrier density and hence cause nonlinear Hall effect in some materials, for instance those undergoing localization-delocalization transition upon increasing magnetization. Following this mechanism, one would expect this effect to be independent of the orientation of magnetic field. This type of isotropic nonlinear Hall effect has been observed previously [e.g. in EuB_6 , Zhang et al., Phys. Rev. Lett. **103**, 106602 (2009)]. As shown in Fig. S10, the nonlinear part of the Hall resistance, R_{NL} , observed in $(\text{Bi,Mn})_2\text{Se}_3$ thin films varies significantly when the field tilting angle is changed. Fig. S10 also shows that the angular dependence of R_{NL} is quite different for two samples with different Mn-doping levels. Such a difference can be attributed to different characteristics of magnetic anisotropy in the two samples and is hard to explain with the independent Mn spins polarized by the magnetic field.

Nevertheless, we agree with the referee that a lot of care must be taken when interpreting the small Hall resistance component obtained by subtracting a linear background. Although the data analyses were not described in sufficient detail in the previous manuscript, we have been cautious in the data analysis throughout this work.

For instance, we have carefully considered how to make other possible sources of the nonlinear Hall effect, such as the multiple-band transport, negligible in the extracted Hall resistances, and how to rule out a paramagnetic phase at low temperatures. Please see Supplementary Figs. S1-S10 and Notes 1-3 for details.

The shape of the AHE vs. field curves don't make much sense at intermediate Mn concentrations, and this is I believe a hint at the uncertainty of the interpretation. I recommend that the authors be more forthright about the uncertainties in the interpretation, and explain in more detail, the choices they have made in separating the Hall conductivity into ordinary and anomalous Hall contributions. The data are interesting and extensive, and do address an important puzzle. The introduction contains a useful summary of relevant literature. I recommend that a MS with a carefully reconsidered analysis which points readers to all the danger points be accepted for publication.

Reply: The shape of the AHE curves at intermediate Mn concentrations can be understood as a crossover between the low doping and high doping samples. In the low and high doping samples, the AHE signals are dominated by the positive component from the bulk and the negative component from the surface, respectively. These two types of AHE components have different magnetic anisotropies and thus different saturation fields. In the samples with intermediate doping levels, the two types of AHE components coexist and lead to the kink structure developed in the low field region. In the revised Figs. 2-4, we have added some curves for the two AHE components separated from the total AH resistances. In addition, the uncertainties arising from the nonlinearity in the ordinal Hall effect are now described in the Supplementary Information (See Note 1 and related figures).

Once again, we thank reviewer #2 for carefully reviewing our manuscript and making many valuable suggestions. We hope the revised manuscript, along with the Supplemental Information, can satisfactorily address all concerns of the reviewer.

Reviewer #3

Comments: The manuscript titled by “Two-component anomalous Hall effect in a magnetically doped topological insulator”, reports the observation of abnormal anomalous Hall (AH) effect in Mn doped Bi₂Se₃ films. The authors have carried out detail investigation of magneto-transport measurement in a series of 10 quintuple-layer (QL) thick (Bi_{1-x}Mn_x)₂Se₃ film with Mn concentration ranging from x = 0 to x = 0.088. The Hall effect is studied as the function of Mn concentration, and of carrier concentration (Fermi level) which is controlled by gating. The paper reports that the

sign of AH resistances can be changed from positive to negative by the Mn concentration, and these two types of AH resistances are found to coexist in the crossover regime. Such a two-component AH effect and the sign reversal can also be obtained by lowering the chemical potential in the samples with low Mn-doping levels. Based on their different dependences on the gate voltage and magnetic field, the authors assigned the positive and negative AH components to the bulk and surface states, respectively. The separation of the two AH components in the resistance data is introduced and the mechanism of these observation are also discussed. The topic of paper is very interesting, and the data are impressive. However, I cannot recommend the publication of the paper as its current form because of two major issues.

Reply: We thank the reviewer for carefully reviewing our manuscript. In the following, we address the comments made by the reviewer.

1. There is no temperature dependent AH resistance data and magnetization data presented in the paper. Thus, it is hard to conclude the observed curvature in the Hall data is due to the magnetization in the films. Clearly, it is well known that there are two kinds of carriers (bulk and surface) in the Bi_2Se_3 films, and these carriers also have different mobility. The Mn concentration and gating can tune the ratio of these two kinds of carriers; therefore, applying a single carrier model (linear Hall resistance) for normal Hall resistance is unjustified.

Reply: We have added two figures in the Supplemental Information to present the temperature dependence data. Figs. S2 and S3 show that the nonlinear part of the Hall resistance of the Mn-doped Bi_2Se_3 thin films has a very strong dependence on temperature, in contrast the nearly temperature-independence in the undoped Bi_2Se_3 sample. Because of the low carrier mobilities in the Bi_2Se_3 thin films grown on SrTiO_3 , the coexistence of multiple types of carriers only leads to negligible amount of deviation from the linear Hall effect in magnetic fields up to 5 T, as shown in Fig. S1. The carrier mobilities in the $(\text{Bi,Mn})_2\text{Se}_3$ samples are either comparable or much lower than those in the undoped Bi_2Se_3 samples, so the multiple-band transport does not introduce substantial error in the extracted AH resistances for the magnetic fields relevant to the bulk and surface magnetism. The validity of the assumption of the linear ordinary Hall effect has been confirmed with the measurements at temperatures both lower and higher than the Curie temperature. Please see Supplementary Note 1 and Figs. S2 & S3 for details.

2. Does the observed magnetization curves have hysteresis loop? Does AH resistance data have hysteresis loop? Are two of them comparable? Although the paper is focusing on high field data, the authors should establish the basic properties of AH resistance as the starting point.

Reply: No clear hysteresis loop could be detected in either magnetization or AHE measurements, as shown in Figs. S9. However, our experiment does not rule out small hysteresis loops with coercive fields on the order of Gauss because of the flux trapping effect in the superconducting magnet. The absence of clear hysteresis should not be viewed as the evidence for the lack of ferromagnetic order, since similar behavior also appears in many soft ferromagnetic materials with T_C above room temperature, such as YIG and BaFe₁₂O₁₉. Moreover, the overall characteristics of the in-plane magnetization of our (Bi,Mn)₂Se₃ thin films are consistent with those reported in Refs. [17-20]. For low Mn-doping samples (e.g. x=0.018, Fig. S2), in which the AHE is dominated by the bulk carriers, the temperature dependences of AH resistances are also in agreement with the out-of-plane bulk magnetization measurements reported in Ref. [20]. The above information was left out in the previous manuscript because we worried about that too many technical details would distract readers from our main message, namely the observation of the two-component AHE.

In addition to the justification of AH data, there are some minor technical issues the authors should also consider when they revised the data.

1. The methods of the separation of two AH components are illustrated in Figs 7 and 8. The second method of the separation is not well described in Figure. Fig. 8 only serves as results for validity. Could the authors produce similar illustration as Fig. 7.

Reply: We thank the reviewer for pointing this out. Detailed illustration of the second method for separating the two AH components is now given in Fig. S6 in the Supplemental Information. Additional comparison of the two methods are also presented with the data from another sample (Fig. S8).

2. The mobility data, both as function of Mn concentration and gating voltage, should be provided.

Reply: We have added the mobility data in Figs. 3(b) and 4(b), as well as in Table 1.

3. The mechanism of opposite sign for bulk and surface AH resistance is not well explained.

Reply: We have added some sentences discussing opposite signs for the surface and bulk AH resistances in the main text. Please see the list of changes for details.

4. The different behavior of surface AH conductivity as function of sheet conductivity (Fig. 5c and Fig. 6c) is very interesting. Is this a general rule, bulk AH conductivity is monotonically increasing as sheet conductivity (Fig 5b and Fig 6b)? On the other hand, surface AH conductivity has a highest point around $5-6 e^2/h$, hasn't it?

Reply: We appreciate the reviewer for keenly noticing a key point of our experimental results. The trend summarized above does appear to be general. Measurements of all our samples seem to be consistent with them. Fig. S8 shows the conductivity dependences of the two AH components for sample D ($x=0.024$), in which the surface AH conductivity has a clear maximum at $\sigma_{xx} \sim 5e^2/h$.

5. Have the authors carried out (field orientation) angular dependent of magneto-transport measurement to separate the conductivity of surface and bulk effect?

Reply: We have carried out the measurements in the tilted magnetic fields. Supplementary Fig. S10 shows the results of two samples. The data indeed further confirm different characteristics for the two AH components. Detailed quantitative analysis is, however, difficult due to complicated competition between the surface and bulk magnetizations in the 10 nm thick thin films. Nevertheless, the tilted-field measurements can help to rule out a mechanism proposed by reviewer #2, namely the change in the ordinary Hall effect caused by the alignment of nearly independent spins by the external magnetic field. Further work on the tilted field measurements may unveil fascinating novel spin structures in this unique magnetic system.

List of Changes:

1. Fig. 1, caption: a sentence is added to refer to a detailed discussion of the magnetic order given in the Supplementary Information.
2. Fig. 2: In the plots for the samples with high Mn concentrations ($x=0.074$ & 0.088), the magnetic field range is expanded to $-9 \sim +9$ T. The curves for the two separated AH components are added in four panels ($x>0.02$). The figure caption is updated accordingly.
3. Page 5, paragraph 1, last 7 lines: A clearer description of the data shown in Fig. 2 is given.
4. Fig. 3: (b) The mobility data are added; (d) The separated positive and negative AH components are shown for two gate voltages (-30 & -210 V). The caption is updated.
5. Fig. 4: (b) The mobility data are added; (d) Separated positive and negative AH components are plotted for two gate voltages (-210 & $+210$ V). The caption is updated.
6. Fig. 6: The AH conductivity values are updated with the Hall coefficient values extracted with a broader fit range (5-9 T). This help to minimize the influence of the unsaturated magnetization and thus improve the data reliability. The basic trend of the obtained AH conductivities is not affected.
7. Table 1: The mobility values are added for all samples.
8. Page 9, lines 1-3: One sentence is added to discuss the sign of the positive AH

component.

9. Page 10, lines 2-8: Three sentences are added to discuss the positive and negative signs of the AH resistances observed in magnetically doped TIs.
10. Page 11, last 5 lines: One sentence is added to discuss the possible novel spin structures emerging from the competition between the surface and bulk magnetic orders.
11. Methods Section: the description of the separation of the two AH components is moved to Supplementary Note 2. Additional figures to illustrate the two separation methods and their equivalence are also provided in the Supplementary Information.
12. Pages 17-18: Five new references are added: Refs. 36, 37, 38, 42, 43.
13. Supplementary Information is added to this submission. It contains 10 figures and 3 technical notes on the nonlinear Hall effect, the AH data analysis methods and discussion of the magnetic orders in the $(\text{Bi,Mn})_2\text{Se}_3$ thin films.

REVIEWERS' COMMENTS:

Reviewer #1 (Remarks to the Author):

----report-----

I have examined the revised manuscript, the answers to the reviewer comments, and the supplementary material.

1. The other reviewers shared my reservation towards the extraction of the anomalous Hall effect (AHE) signal. The supplementary information provided with the resubmitted manuscript is indeed helpful here.
2. I had previously asked about the magnetic anisotropy. The authors confirm that the magnetic anisotropy is in plane. This is for the bulk. And they refer to some out of plane magnetization reported in the literature for the surface.

On the other hand, it is clear that the AHE is by at least an order of magnitude smaller than what is found in other magnetically doped topological insulator systems. For example in ref. 9 (Checkelsky et al.).

This means first of all that the small size of the AHE and the (bulk) magnetic properties are not inconsistent.

3. I do not understand how the different anisotropies of bulk and surface can be present, and the authors also express this concern (the thickness is only 10 nm). The simplest explanation (which is not at all discussed in the manuscript) is still an inhomogeneous sample. However, I acknowledge that the gating dependence indicates that the surface may be one of the two components of the AHE with opposite sign. It is indeed remarkable that comparing figs. 2 and 3 the dependence of the AHE on the Mn concentration and on the gate voltage are very similar. The authors argue that their Mn concentration dependence is consistent with the ARPES data of ref. 20. This means that the chemical potential changes by almost 100 meV between pure Bi₂Se₃ and Mn doping of $x=0.08$. This is not enough to get the sample in the topological transport regime but it may change the ratio of surface to bulk transport according to the sketch of the chemical potential in fig. 1f. The carrier concentration dependence indicates that the gating drives the chemical potential somewhat further than the Mn concentration does. This is also consistent with the previous results of ref. 9 for a different system.

In conclusion, I am not convinced of the interpretation but I cannot offer a better one in view of the presented data. I am also not convinced of the general interest justifying Nature Communications. If, as it seems, the other reviewers find the results interesting enough, I will not object to publication in Nature Communications.

----end-----

Reviewer #2 (Remarks to the Author):

I still regard the bulk/surface separation of the AHE described in this paper to be uncertain, like many other bulk/surface separations in the literature. Still the data is extensive and interesting and now more carefully described. I recommend acceptance of this version of the MS which describes timely

research that will be valued by readers.

Reviewer #3 (Remarks to the Author):

The authors have response all comments from three reviewers. I am satisfied with their response and they certainly clarified the issues regarding tow-component AHE in BiMnSe. I would like to suggest the publication of the paper as is in Nature Communications. A minor issue, which the authors might consider, is that the paper does not refer to the supplementary figures at all.

Response to the 2nd round of reviewer comments

Reviewer #1:

Comments:

I have examined the revised manuscript, the answers to the reviewer comments, and the supplementary material.

1. The other reviewers shared my reservation towards the extraction of the anomalous Hall effect (AHE) signal. The supplementary information provided with the resubmitted manuscript is indeed helpful here.

2. I had previously asked about the magnetic anisotropy. The authors confirm that the magnetic anisotropy is in plane. This is for the bulk. And they refer to some out of plane magnetization reported in the literature for the surface.

On the other hand, it is clear that the AHE is by at least an order of magnitude smaller than what is found in other magnetically doped topological insulator systems. For example in ref. 9 (Checkelsky et al.). This means first of all that the small size of the AHE and the (bulk) magnetic properties are not inconsistent.

Reply:

We are glad that the reviewer found the revised manuscript and the Supplemental Information useful in helping him/her to understand several key points of our experimental results.

Comment:

3. I do not understand how the different anisotropies of bulk and surface can be present, and the authors also express this concern (the thickness is only 10 nm). The simplest explanation (which is not at all discussed in the manuscript) is still an inhomogeneous sample. However, I acknowledge that the gating dependence indicates that the surface may be one of the two components of the AHE with opposite sign. It is indeed remarkable that comparing figs. 2 and 3 the dependence of the AHE on the Mn concentration and on the gate voltage are very similar. The authors argue that their Mn concentration dependence is consistent with the ARPES data of ref. 20. This means that the chemical potential changes by almost 100 meV between pure Bi₂Se₃ and Mn doping of $x=0.08$. This is not enough to get the sample in the topological transport regime but it may change the ratio of surface to bulk transport according to the sketch of the chemical potential in fig. 1f. The carrier concentration dependence indicates that the gating drives the chemical potential somewhat further than the Mn concentration does. This is also consistent with the previous results of ref. 9 for a different system.

Reply:

We agree with the reviewer that there is a difference between the Mn-doping and the gating with respect to their influences on the chemical potential. Without gating, even the doping up to $x=0.08$ is not sufficient to get the sample into the topological transport regime. This has been clearly illustrated in the band diagram in Fig. 1f. We

also agree with the reviewer that the overall similar trend displayed in the gate-voltage and the doping-concentration dependences of the AHE can be used to rule out the possibility of the two AHE components arising from the sample inhomogeneity. In addition, during the growth of all $(\text{Bi,Mn})_2\text{Se}_3$ thin films, we maintained a low substrate temperature to prevent the formation of secondary phases. According to a previous structural study [19], no Mn-containing secondary phases formed in the $(\text{Bi}_{1-x}\text{Mn}_x)_2\text{Se}_3$ thin film with the Mn concentrations up to at least $x=0.075$.

Regarding the presence of different magnetic anisotropies of the bulk and surface in the same sample, we would like to make the following comments. As we pointed out in the manuscript, the evidence for the in-plane anisotropy in the bulk of $(\text{Bi,Mn})_2\text{Se}_3$ has been overwhelming [17,18,22]. Previous theories and experiments also support the perpendicular out-of-plane anisotropy for the surface states in magnetically doped TIs [26,30,31,9,13,14,36,37]. To be precise, the question which needs to be addressed for the $(\text{Bi,Mn})_2\text{Se}_3$ thin films is whether the surface states can have perpendicular anisotropy under the influence of the in-plane magnetic ordering in the bulk. Prior to our work, the only experiment on probing the surface state magnetism in Mn-doped Bi_2Se_3 thin films is the XMCD measurement reported in Ref. [20], which shows that there is at least an out-of-plane magnetization component for the surface states. In our work, the high saturation field of the positive AHE component is consistent with the in-plane anisotropy of the bulk magnetization. In contrast, the much lower saturation field of the negative component and its drastically different gate-voltage dependence can only be attributed to the surface magnetization. As we pointed out in the manuscript, it is very likely that the spin interactions between the surface and bulk electrons in the 10 nm thick $(\text{Bi,Mn})_2\text{Se}_3$ thin films cannot be neglected, and they could lead to interesting complex spin structures such as spin canting, and non-collinear magnetic order. The tilted field measurements shown in Supplementary Figure 10 seem to provide a piece of preliminary evidence for such interactions. There is no doubt that further work is needed to elucidate the nature of magnetic ordering in magnetically doped TIs, as stated in the last paragraph of the revised manuscript.

Comment:

In conclusion, I am not convinced of the interpretation, but I cannot offer a better one in view of the presented data. I am also not convinced of the general interest justifying Nature Communications. If, as it seems, the other reviewers find the results interesting enough, I will not object to publication in Nature Communications.

Reply:

We thank the reviewer for making a lot of efforts in reviewing our manuscript. His/her criticisms are valuable in improving the quality of the manuscript. As we stated before, the two most important findings of our work are: (1) the first observation of the elusive AHE in the Mn-doped Bi_2Se_3 thin films, in which the ground state spin properties are still controversial, (2) the first observation of a two-component AHE in magnetically doped TIs, and possibly in any magnetic material (of course, not including artificial bilayer or multilayer structures). Such a two-component AHE could in

principle also exist other materials and a deep understanding of this effect could gain deep insight into the magnetic ordering in complex magnetic systems. As we pointed out above, the competition between the two AHE components could result in novel spin structures. Therefore, we believe that our manuscript will be attractive to readers from communities of magnetism, electron transport, topological materials, as well as magnetic semiconductors.

Reviewer #2 :

Comment:

I still regard the bulk/surface separation of the AHE described in this paper to be uncertain, like many other bulk/surface separations in the literature. Still the data is extensive and interesting and now more carefully described. I recommend acceptance of this version of the MS which describes timely research that will be valued by readers.

Reply:

We thank the referee for recommending our manuscript to be published. As to uncertainty in the assignment of the two AHE components to the surface and bulk magnetizations, we have done everything we could to exclude all other possibilities. Of course, further work is necessary to elucidate the nature of magnetic ordering in $(\text{Bi,Mn})_2\text{Se}_3$ thin films or more generally in magnetically doped TIs, in particular with microscopic probes that are sensitive to surface spin structures.

Reviewer #3:

Comment:

The authors have response all comments from three reviewers. I am satisfied with their response and they certainly clarified the issues regarding two-component AHE in BiMnSe. I would like to suggest the publication of the paper as is in Nature Communications. A minor issue, which the authors might consider, is that the paper does not refer to the supplementary figures at all.

Reply:

We thank the referee for supporting the publication of our manuscript. We also thank him/her for pointing out the insufficient references to the Supplementary Figures in the main text. We have fixed this problem in the revised manuscript.